# Simulation and Analysis of an FMCW Radar against the UWB EMP Coupling Responses on the Wires

**DOI:** 10.3390/s22124641

**Published:** 2022-06-20

**Authors:** Kaibai Chen, Shaohua Liu, Min Gao, Xiaodong Zhou

**Affiliations:** 1Department of Missile Engineering, Army Engineering University, Shijiazhuang 050003, China; ckbguessg@163.com; 2Beijing Institute of Systems Engineering and Information Control, Beijing 100020, China; lshmarch@163.com; 3Department of Ammunition Engineering, Army Engineering University, Shijiazhuang 050003, China; zxdoec@163.com

**Keywords:** ultra-wideband electromagnetic pulse, FMCW radar, joint simulation, wire coupling responses

## Abstract

An ultra-wideband electromagnetic pulse (UWB EMP) can be coupled to an FMCW system through metal wires, causing electronic equipment disturbance or damage. In this paper, a hybrid model is proposed to carry out the interference analysis of UWB EMP coupling responses on the wires to the FMCW radar. First, a field simulation model of the radar is constructed and the wire coupling responses are calculated. Then, the responses are injected into an FMCW circuit model via data format modification. Finally, we use the FFT transform to identify the spectral peak of the intermediate frequency (IF) output signal, which corresponds to the radar’s detection range. The simulation results show that the type of metal wire has the greatest influence on the amplitude of coupling responses. The spectral peak of the IF output changes to the wrong frequency with the increase of injection power. Applying interference at the end of the circuit can more effectively interfere with the detection of the radar. The investigation provides a theoretical basis for the electromagnetic protection design of the radar.

## 1. Introduction

The frequency modulated continuous wave (FMCW) radar uses modulated electromagnetic waves to sense targets. It has been widely applied in military and civilian fields because of its high ranging accuracy, good anti-jamming performance, and strong range selection ability [1,2]. However, the electromagnetic environment faced by FMCW radars is becoming more and more complex with the development of electronic technology, which makes researchers pay attention to the influence of UWB EMP on FMCW radars [3,4].

UWB EMP usually refers to the electromagnetic environment with a field strength of more than 1000 V/M and a frequency range of several kHz to 300 GHz [5]. It can be coupled into the electronic system through antennas, wire connections, apertures, or holes and results in induced voltage, which will cause serious damage to the electronic system. Three methods are typically used in electromagnetic interference (EMI) analysis: the analytic method, the experimental method, and the numerical method. The analytic method is a fast way to calculate the responses in an enclosure equipped with apertures. However, one can hardly derive the correct analytical formula when the research object is irregular [6,7]. In addition, using analytical methods to solve the UWB EMP coupling responses often needs to combine multiple analysis models like the Agrawal model or BLT model. This will affect the accuracy and efficiency of the results [8,9]. The experimental method is the most direct way to investigate the coupling effects of the UWB EMP on an electronic system. Regular electronic systems, such as low-noise amplifiers [10], micro-processors [11,12], PC networks [13], UAV systems [14], GPS systems [15], aircraft systems [16], and radar enclosures [17], have been investigated for susceptibility in the laboratory. By collecting the experimental data, researchers can get the statistical law of an electronic system against the UWB EMP, then provide relevant advice to reinforce the protection, and apply it to engineering practice, like in the electromagnetic protection field. However, the experimental method occupies too many resources, and the high cost is also worthy of attention. In addition, in order to obtain accurate sensitive parts, it is necessary to combine a variety of experimental approaches, such as irradiation tests and injection tests. Therefore, the experimental method is often used for the final verification of electronic equipment. For the auxiliary experimental method, numerical methods have been created and are widely used in the electromagnetic compatibility (EMC) field. The accuracy of numerical methods is much higher than that of analytical methods, and compared with experimental methods, the cost of numerical methods is almost negligible. Among the numerical methods are the three-dimensional full-wave transmission line method (3D-TLM), the finite integral time-domain method (FIT), the method of moments (MoM), the finite element method (FEM), and others [18,19,20,21,22]. However, these numerical methods generally have trouble treating the active circuit, and circuit-based techniques that can handle the active circuit cannot analyze the field coupling responses [23,24,25,26,27]. Although many electronic systems have been investigated in previous work, little work has been reported in the open literature on the FMCW radar. The works [3,4] study the interference law of narrow-band EMP and UWB EMP on IF signal through a radar antenna using a hybrid model. However, the hybrid model only considers the coupling voltages of the radar antenna and does not consider the influence of the metal enclosure on the coupling voltage nor the interference of the wire coupling voltages on the IF signal. In fact, since the operating frequency of the radar is much higher than the spectrum coverage of UWB EMP, the coupling voltage on the radar antenna is much smaller than that on the metal wire. In addition, the UWB EMP signal from the front-door channel has little interference with the radar because the antenna and the filter on the RF circuit board have certain filtering abilities that will attenuate a large part of the coupling energy.

The comparison of existing methods is briefly shown in Table 1. We can see from it that a complete hybrid model is required when assessing the EMI effects of FMCW systems. As a result, this paper proposes a hybrid model which combines the actual structure, the field model, and the circuit model, so that the interference of wire coupling responses induced by UWB EMP to a FMCW radar can be investigated. The conclusion may offer recommendations for radar electromagnetic protection.

This paper is organized as follows: the relevant theory is briefly introduced in Section 2. In Section 3, the hybrid model is proposed. In Section 4, the results and analysis are presented. The discussion is presented in Section 5. Finally, the conclusions are given in Section 6.

## 2. Relevant Theory

### 2.1. The Coupling Theory

Some related theories are introduced in this section. UWB EMP energy can enter electronics through the “front-door” or the “back-door”. The term “front-door” refers to the electromagnetic pulse energy coupled into the radar radio frequency (RF) front-end through receiving antennas or wire connections. When the coupling energy is too large, the RF circuit will be saturated, blocked, or even burned. Due to the high working frequency of the radars, the coupling energy between the antenna and UWB EMP often attenuates significantly. Hence, it is unlikely that the antenna coupling responses will interfere with the radar. However, the wire connections in the radar can be compared to an additional receiving antenna, which may bring on large induced voltage or current and then lead to the radar failure.

Generally, the coupling power of the radar antenna can be calculated by the Friss transmission formula:(1)$$P_{r}=\frac{P_{t}G_{t}G_{r}\lambda^{2}B_{r}}{{\left(4\pi R\right)}^{2}B_{t}L_{e}}$$
where:$P_{r}$ is the coupling power of receiving antenna;$G_{t}$ is the gain of the transmitting antenna;$G_{r}$ is the gain of receiving antenna;λ is the working wavelength of receiving antenna;$B_{r}$ is the bandwidth of receiving antenna;R is the target range between the UWB EMP and the antenna;$B_{t}$ is the bandwidth of the transmitting antenna of UWB EMP;$L_{e}$ is the loss factor in the coupling process.

The Friss transmission formula demonstrates that coupling power is related to numerous characteristics, and it is difficult to quantify precise parameters using analytical or experimental methods, particularly when the wire connections are shielded by the radar’s enclosure.

### 2.2. Ranging Principle of the FMCW Radar

The FMCW radar uses frequency variations between echo signals and transmitting signals to determine the target range. When the transmitting signal contacts the target, an echo signal is produced and accepted by the receiving antenna. The difference in frequency between the signals, including the target range, is output through the mixer, which can mix the echo signal and local oscillator (LO) signal. Then, the radar emits an alert signal when the target range meets the specified condition. Figure 1 depicts the ranging concept, which can be used to describe the process.

Assuming that the radar and the UWB EMP source are on the same horizontal plane and the radar moves in a straight line towards the UWB EMP source, considering the doppler effect, the FMCW radar ranging formula can be described as [28]:(2)$$\Delta f=\frac{4BR}{cT_{M}}+\frac{2f_{0}v}{c}$$
where:Δf is the difference frequency;B is the modulation bandwidth;c is the speed of light;$T_{M}$ is the modulation period;$f_{0}$ is the carrier frequency;v is the velocity of the FMCW radar.

## 3. The Hybrid Model

### 3.1. The Flowchart

The hybrid model is divided into two parts: field analysis and circuit analysis. In the field analysis part, the radar model and the UWB EMP are built in CST Microwave Studio, which is professional in the field of electromagnetic simulation. The wire connections are defined. The coupling responses are computed through the full-wave 3D-TLM, and then it is loaded into the schematic environment to calculate the wire coupling responses. In the circuit analysis part, ADS, which is widely employed in the circuit design, is used to establish the model of the FMCW radar, which contains the transmitter, the target, and the receiver, as well as the key nodes. The results of field analysis can be transferred into circuit analysis by converting the data format. The flowchart of the proposed model is shown in Figure 2.

### 3.2. The Field Model

#### 3.2.1. The Simplified Model of the FMCW Radar

The radar is shielded by a metal enclosure (see Figure 3). The RF board, which is responsible for the signal transmission, is attached at the top of the enclosure. The metal wires extend outward from the enclosure so that the radar can deal with the power supply and the control signals. The internal structure of the radar is shown in Figure 4. The digital signal processing (DSP) board for echo analysis is installed inside the enclosure, and the wires are connected to the corresponding nodes.

The simplified field model of the radar is shown in Figure 5. The dimensional parameters are shown in Figure 6. The field model preserves the majority of the radar structure, including the enclosure, RF board, and DSP board. In order to speed up the simulation, the DSP board is considered as a layer, and the metal wire is placed in the center of the DSP board.

#### 3.2.2. Simulation Setup

In the field analysis, the UWB EMP is assumed as a horizontally incident, vertically polarized plane wave. It propagates along the metal wire parallel to the *x*-axis, and the electric field component vibrates on the *z*-axis, which can maximize the electric field intensity in the radar. The incident direction of UWB EMP is shown in Figure 7, and it can be expressed as:(3)$$x(t)=E_{0}\frac{t-t_{0}}{t_{w}}e^{-4\pi {\left(\frac{t-t_{0}}{t_{w}}\right)}^{2}}$$
where $E_{0}$ is the electric field factor of UWB EMP, $t_{0}$ is the delay time of the pulse, and $t_{w}$ is the width constant of the pulse.

The simulation frequency is 2 GHz, the background property is normal, and the boundary is open. The RF board and DSP board are made of FR-4, which is often used to make multilayer printed circuit boards. The antenna is made of pure copper, and the radar enclosure and isolator are made of aluminum. The time-domain solver is used in the simulation, and the simulation accuracy is −50 dB. Furthermore, each node of the wire has a 50 ohm load, and probes are attached at the ends of the wires to monitor the induced voltage.

### 3.3. The Circuit Model

#### 3.3.1. The Composition of the FMCW Radar

The FMCW radar is made up of the transmitter, power divider, transmitting and receiving antennas, target model, mixer, and IF output, as illustrated in Figure 8. The transmitter can generate FMCW with a specific bandwidth, which is then divided into two parts by the power divider: one is radiated into free space via the transmitting antenna and reflected by the receiving antenna after encountering the target, and the other is sent to the mixer and mixed with the echo signal. The Chebyshev filter in the IF part can screen certain echo signals to calculate the detection range, as well as restrict the power of out-of-band signals and safeguard the IF amplifier. A time delay component, an attenuator, a Doppler shift component, and a radar cross-section (RCS) component are all included in the target model. The time delay component describes the target range and can calculate the echo signal receiving time, the attenuator represents channel loss, the doppler shift component can calculate the frequency shift caused by the moving target, and the RCS component can calculate the echo signal reflection coefficient based on target characteristics.

In the circuit model, the time delay and the *RCS* are expressed as follows:(4)$$TimeDelay=\frac{2R}{c}$$
(5)$$RCS=\sqrt{\frac{T_{c}}{{\left(4\pi R^{2}\right)}^{2}}}$$
where, R is the target range, and $T_{c}$ is the cross-section of the target. Other parameters of the circuit model are listed in Table 2.

#### 3.3.2. The Verification of the Circuit Model

A transient simulator is used in the verification. The stop time is set to 0.5 us and the max time step is set to 0.01 ns. According to formula (2), the target range corresponds to the spectral peak of IF output. As shown in Figure 9, the spectral peaks corresponding to 5 m, 10 m, 15 m, and 20 m are 10 MHz, 20 MHz, 30 MHz, and 40 MHz, respectively, which verifies the correctness of the FMCW behavior model.

## 4. Results and Analysis

### 4.1. Field Simulation Results and Analysis

The properties of the incident wave will influence the coupling responses, such as frequency bandwidth, pulse width, size of the incident wave, polarization, and angle of incidence. The polarization and angle of incidence of UWB EMP are determined in the field analysis in 3.2.2, and Figure 10 illustrates the time-domain and frequency-domain waveforms of the applied UWB EMP. As shown in Figure 10a, the peak value of the electric field intensity is 1 V/m, and the pulse rise time is approximately 0.141 ns. Thanks to the contribution of the rise time, the pulse covers a wide spectrum and its energy is mainly concentrated on 1.6 GHz. The peak of the power spectrum can reach 1.004 dBm.

In this part, we have simulated six cases in which we’re concerned with the type, length, radius, curvature, number, and line spacing of the wires (see Table 3). A variable is set in each case so that the impacts of the factor can be observed.

#### 4.1.1. The Impact of the Wire Types

In case 1, the coupling responses of a single wire (LYFY-0 qmm50), a twisted wire (UTP LIFY-1 qmm), and a coaxial wire (RG-58) are simulated. The wires come from the model library of CST, and the actual geometry is shown in Figure 11.

It is noted that the single wire and the twisted wire are composed of the conductor and the insulator, while the coaxial wire is composed of the conductor, insulator, and the screen. The parameters are shown in Table 4.

The electromagnetic field structure is illustrated in Figure 12. As we can see from the circle, the electric fields near the metal wire are stronger, which means that the metal wire has a certain attraction to the UWB EMP.

As shown in Figure 13, waveform oscillation is formed when the wire coupling responses bounce back and forth because of the reflection. The peak voltages of the three wires are 2.200 × 10^−4^ V, 1.363 × 10^−4^ V, and 1.017 × 10^−8^ V, respectively. Due to the grounded shielding, the coupling responses of a coaxial wire are greatly attenuated. This means that the type of wire connection has a significant impact on the SE of the radar.

#### 4.1.2. The Impact of Wire Lengths

Case 2 simulates the coupling responses of a single wire of various lengths. The lengths are 6 mm, 8 mm, 10 mm, 12 mm, and 14 mm, respectively. As can be seen in Figure 14, the coupling responses of longer wires are higher, though the waveforms are essentially the same. Figure 15 shows the relationship between peak time and peak voltage with wire lengths. As the length of a single wire increases, the peak time of coupling responses delays gradually and the peak voltage changes linearly, indicating that it is preferable to use short wires in radar design to improve shielding effectiveness.

#### 4.1.3. The Impact of the Conductor Radius

In case 3, the coupling responses of different radii of a single wire are simulated. Considering the actual application, the radii are 0.1 mm, 0.5 mm, and 1 mm, respectively. As can be seen in Figure 16, the wire coupling responses come to the peak at the same time, which means that the radius of the conductor does not affect the peak time. In addition, the difference of 10 times the conductor radius can only get the corresponding peak value of 1.31 times, which shows that the impact of the conductor radius on coupling responses is limited.

#### 4.1.4. The Impact of the Curvature

In case 4, the coupling responses of different curvatures of a single wire are simulated. The geometry of the wires is shown in Figure 17. The curvature of the conductor does not affect the peak time either, as seen in Figure 18. When the wire length is constant, the curvature of the wire affects the coupling responses. The straight wire has the highest peak value, and the larger the curvature of the line, the lower the peak value of the coupling responses.

#### 4.1.5. The Impact of the Wire Numbers

In Case 5, the single wires are used as the simulation object. The length of the wires is 10 mm, the line spacing between them is 0.5 mm, and the wire coupling responses of different numbers are calculated (see Figure 19). As can be seen in Figure 20, it is noted that the greater the number of wires in the radar enclosure, the lower the coupling responses of the metal wires because the direction of the crosstalk voltage existing in the wires is opposite to the coupling responses. When the number of wires increases, the crosstalk voltage also increases, resulting in a decrease in wire coupling responses.

#### 4.1.6. The Impact of Line Spacing

In case 6, the coupling responses at different distances among the wires are simulated (see Figure 21). In this case, three straight wires are put in the center of the enclosure, and the responses of the middle wire are monitored. The results can be seen in Figure 22. As in case 5, the crosstalk noise between multiple wires remains and affects the wire coupling responses. With the increase in the distance between wires, the crosstalk noise decreases accordingly, resulting in increases in wire coupling responses.

Summarizing the field simulation results, we can know that in the above six cases, the type of wire has the greatest impact on the wire responses. Therefore, in the circuit analysis, we use the coupling responses of a single wire and a coaxial wire as the EMI sources, respectively, and carry out the injection simulation test in the FMCW behavior model to further analyze the interference law of UWB EMP on the FMCW radar.

### 4.2. Circuit Simulation Results and Analysis

The schematic of the circuit analysis for the EMI injection is illustrated in Figure 23. The coupling responses of a single wire (see Figure 13) are employed as the EMI source. It is noted that the wire coupling responses obtained in the field analysis cannot be used directly in the circuit analysis. Thus, the data format conversion is necessary. To realize the conversion, the coupling responses obtained in the field analysis are exported in ASCII format first, then the start character of the exported data is deleted, and a new start character is added:

BEGIN TIMEDATA

# T (NSEC V R 50)

% time voltage

Finally, the wire coupling responses obtained in the field analysis are converted to TIM format, which can be loaded into the circuit analysis.

In consideration of that, the incident wave does not couple into the structure of the passive and active lumped elements at frequencies whose wavelength is significantly larger than the physical size of these components, so the parameters of these elements are not changed as a result of the incident wave. Therefore, in the behavioral simulation, node 5 and node 6 are considered as the injection nodes, and we assume that the electronics remain in linear operation. In addition, all simulation settings are the same as those in Section 3.3.2.

(1)Injection of EMI at node 5;(2)Injection of EMI at node 6;(3)Injection of EMI at nodes 5 and 6.

#### 4.2.1. Single-Point Injection

In the circuit analysis, the wire coupling responses of a single wire are applied. When the target range is 20 m, Figure 24 depicts the interference law of the injection at node 5, where different line types represent different injected EMI amplifications, and the orange pattern means the interfered frequencies. The EMI injection can affect the IF of the echo signal, and as the injection amplitude increases, the peak value of IF gradually changes from 40 MHz to 8 MHz, indicating that the radar has been disrupted and triggered the false control signal. Furthermore, at 40 MHz, the IF output signal’s amplitude has been increased by nearly 11 dBm. This is because the EMI energy in the filter passband has also increased. This is due to an increase in EMI energy in the filter’s passband. The tendency is in agreement with the findings conducted by [4].

A similar result can be obtained when the injection point is changed to node 6. Figure 25 shows that the energy at 0 MHz (green pattern) and 10 MHz (blue pattern) rapidly increases, and the amplification is essentially equal to the amplification of EMI. This means that, without the filter’s function, EMI interference has become more apparent. Furthermore, the peak value at 10 MHz rises, perhaps leading the radar to misjudge the distance.

#### 4.2.2. Two-Point Injection

In Figure 26, the injection points are node 5 and node 6, and the injection amplification of EMI is the same. Compared with Figure 24 and Figure 25, the difference in amplification at 40 MHz is nearly 16 dBm, approximately equal to the superposition value of single-point injection in node 5 and node 6, and the interference frequencies are 0 MHz (green pattern), 8 MHz (orange pattern), and 10 MHz (blue pattern), which are the same as the results of single-point injection.

#### 4.2.3. The Analysis of EMI Amplification and Peak Voltage of IF Output

Table 5 shows the minimum amplification of EMI sources. It can be seen from Figure 26 that the power required for the two-point injection is between the power required for node 5 injection and that required for node 6 injection. From the statistical data, we can learn that the EMI amplification is negatively correlated with the target range. The minimum EMI amplification of the coaxial wire coupling responses increases by approximately 80 dB. This is consistent with the field simulation results in order of magnitude (see Figure 13).

Table 6 displays the peak value of IF output of a single wire. We can see that the increase of interference power of UWB EMP raises the peak voltage of the IF output signal. When the target range is 20 m, the peak voltages of IF output of the three injection conditions are all below 1 V. However, the peak voltage of IF output of two-point injection is 11.5 V when the target range is 5 m, the excessive transient voltage will interfere with or destroy the radar’s DSP, affecting its functional state. As a result, the above issues shall be taken into account in the EMI protection design [11,12]. The simulation findings show that utilizing a coaxial wire with a shielding effect rather than a single wire can protect the radar better.

## 5. Discussion

Based on the joint simulation of CST Microwave Studio and ADS, this paper establishes the field model of the actual radar and studies the IF interference of the FMCW radar in the circuit analysis. In this paper, a complete radar enclosure model is established to improve the accuracy of the calculation. Considering the high working frequency of the radar, the UWB EMP interference energy entering from the front-door channel is actually very weak, so this paper mainly analyzes the wire coupling responses.

We have investigated six factors that could influence the wire coupling responses in the field simulation. From the simulation results, it can be concluded that shortening the length of wires inside the radar, using thinner wires, increasing wire curvature reasonably, and reducing the number of wires will decrease the wire coupling responses. Besides, multiple wires in the radar should be tied together as much as possible. The type of metal wire has the greatest influence on the coupling responses. For example, using a coaxial wire can reduce the amplitude of the interference signal to 1/20,000th of that of a single wire.

In the circuit simulation, we find that the spectral peak of the IF output signal will change with different injection nodes. When the injection power reaches a certain threshold, the spectral peak of the IF output signal changes to an unwanted frequency. The findings are consistent with the results in [4]. When the circuit components are working in the linear area, the interfered spectral peak generated by two-point injection is a synthesis of that generated by single-point injection.

It can be seen from Table 5 that EMI amplification is negatively correlated with the target range. The required EMI power is raised in short-range detection because of the lower energy attenuation of the echo signal, which means that a shorter detection range can bring better anti-UWB EMP capability to the radar. Besides, the interference energy required for node 6 injection is less than that for node 5 injection. As a result, the transmission of IF signals should be better protected. The wire connections can be changed to coaxial wires, or the lengths can be shortened to reduce the coupling responses.

In the case of two-point injection, the EMI injected at node 5 increases the overall power of the filtered signal. Therefore, the EMI power required at node 6 increases accordingly, which will further deteriorate the working environment of the DSP because the peak voltage of the IF output is also increasing. If the peak voltage exceeds the withstand voltage of the DSP, it will lead to digital failure of the FMCW radar. The conclusion is based on the premise that the injection power of these two points is consistent.

Compared with the previous research, the novelty of this paper is to establish a complete behavioral model of the FMCW system and use the hybrid model to study the anti-EMI ability of the radar. Compared with the existing methods, the proposed method can deal with more complex structures and the anti-interference analysis of active circuits. Furthermore, the difference in the effect phenomenon produced by different injection points can be observed intuitively, which helps us take targeted EMI protection measures. The hybrid method can be applied to any electronic device in theory. Of course, the premise is to establish an accurate field model and circuit model.

## 6. Conclusions

In this paper, a hybrid model is proposed to investigate the UWB EMP coupling responses on the wires inside a FMCW radar. According to the simulation, the following conclusions can be obtained:(1)UWB EMP may cause FMCW radar damage via wire coupling responses. The type, length, radius, curvature, number, and line spacing of metal wires will have an impact on the wire coupling responses, but the type of wire connections has the greatest impact on them. Under the same state, the coupling responses on a single wire are 22,000 times those on a coaxial wire. Due to the existence of crosstalk voltage between wires, it is best to bundle the multiple metal wires in the radar.(2)The EMI source can actually affect the spectral peak of the IF output. The main reason is that the UWB EMP covers a wide spectrum range. During high-power injection, it will lead to the spectral peak shift of the IF output signal, resulting in the false control signal output by the radar.(3)To effectively protect the radar from UWB EMP interference, the protection at the IF signal output shall be strengthened, a transmission wire with better shielding performance shall be selected, and the length of the transmission wire shall be shortened as far as possible.

However, the proposed model is based on behavioral simulation and does not take the nonlinear effects of devices into account. Therefore, in future research, we will improve the hybrid model and compare the results with real measurement results. The nonlinear effects of the FMCW radar should also be investigated.

## Figures and Tables

**Figure 1 sensors-22-04641-f001:**
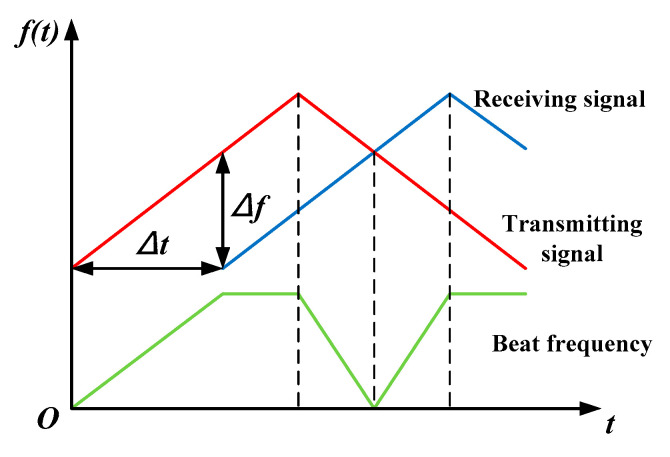
The brief ranging principle of the FMCW radar.

**Figure 2 sensors-22-04641-f002:**
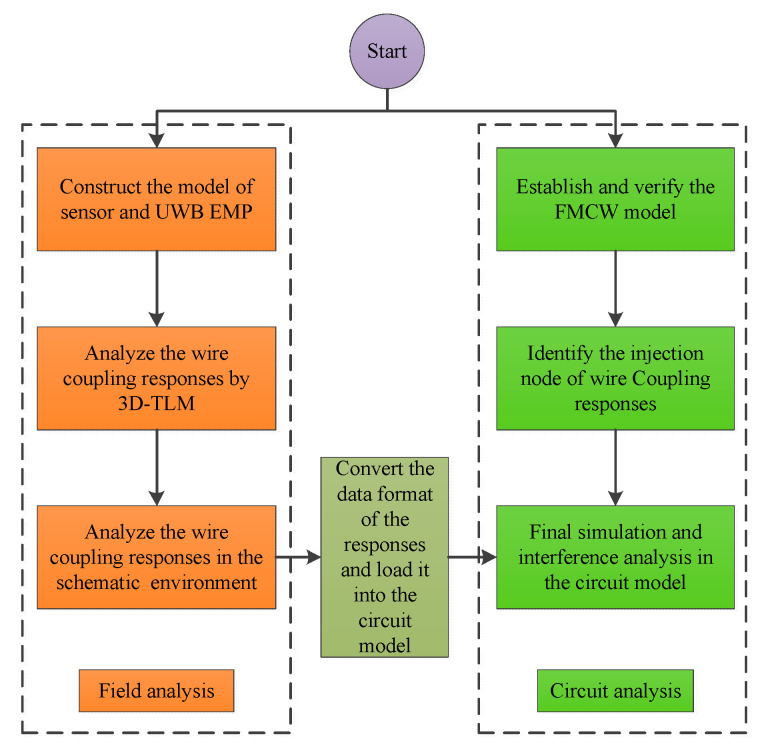
The flowchart of the proposed hybrid model.

**Figure 3 sensors-22-04641-f003:**
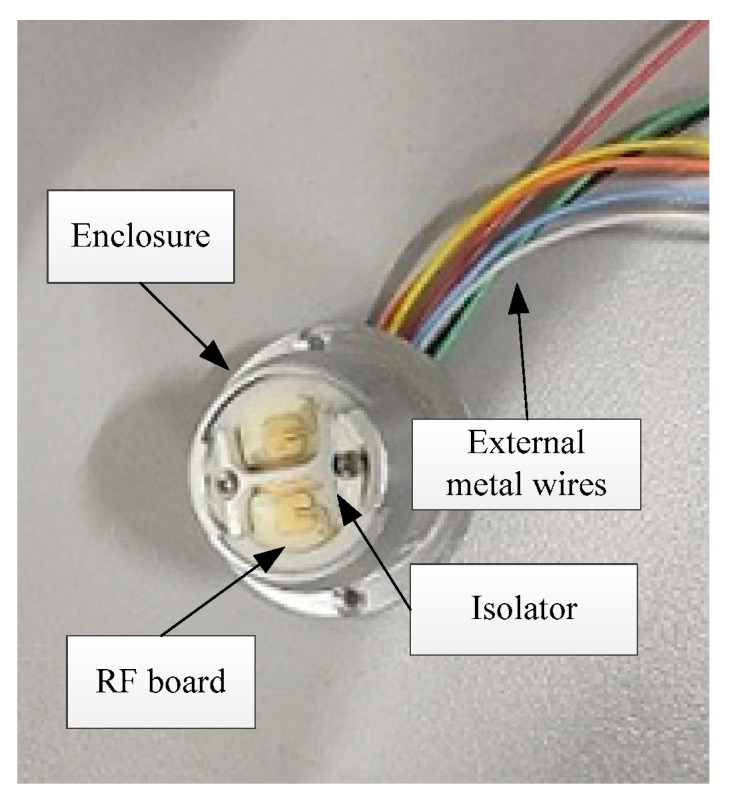
The external structure of the radar.

**Figure 4 sensors-22-04641-f004:**
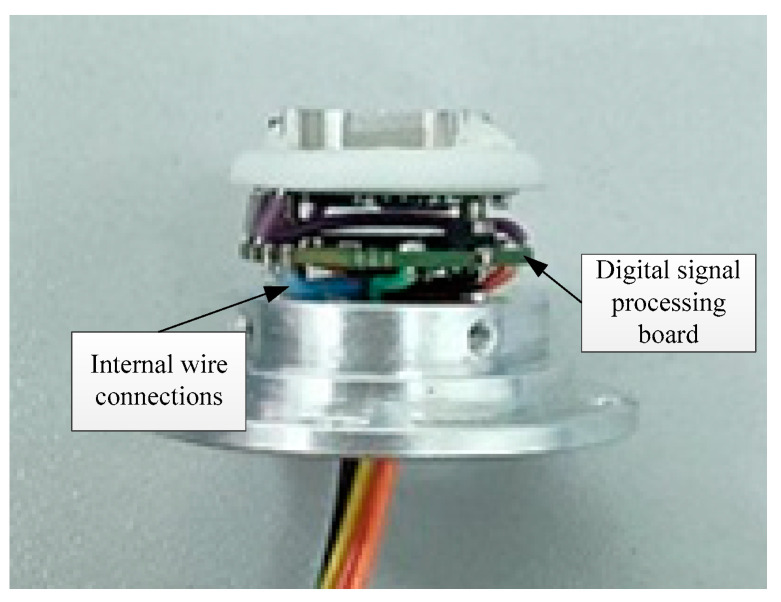
The internal structure of the radar.

**Figure 5 sensors-22-04641-f005:**
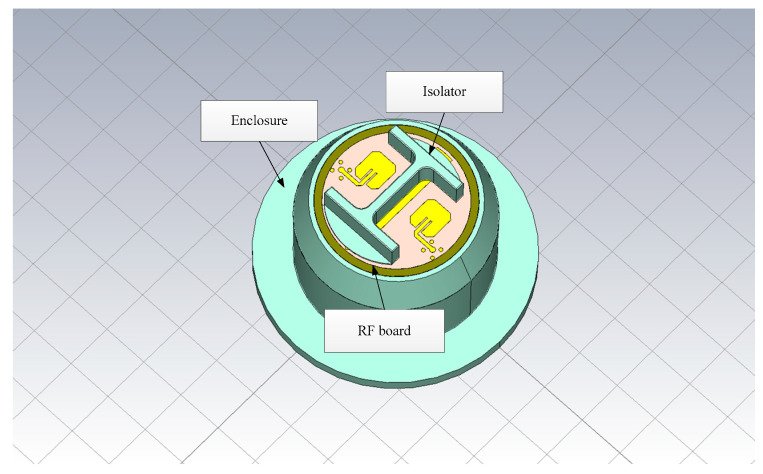
The external structure of the field model.

**Figure 6 sensors-22-04641-f006:**
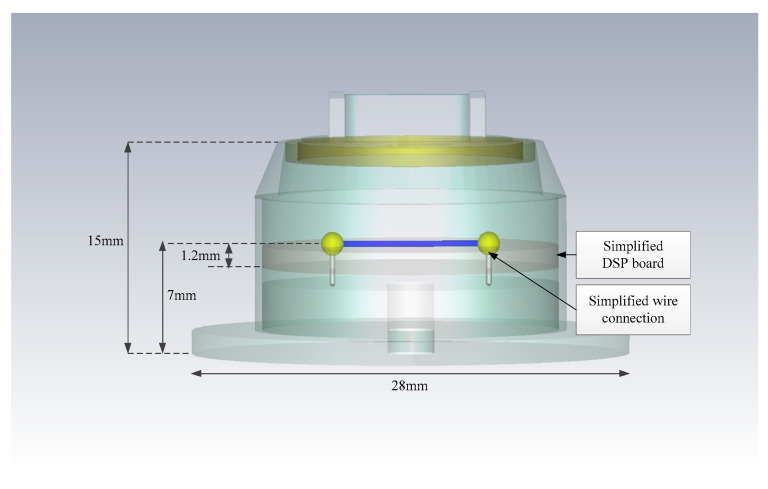
The internal structure of the field model.

**Figure 7 sensors-22-04641-f007:**
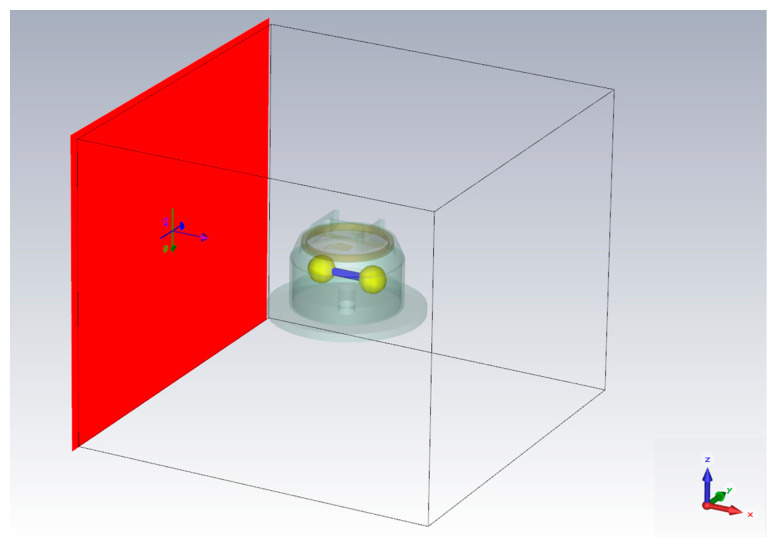
The incident direction of UWB EMP.

**Figure 8 sensors-22-04641-f008:**
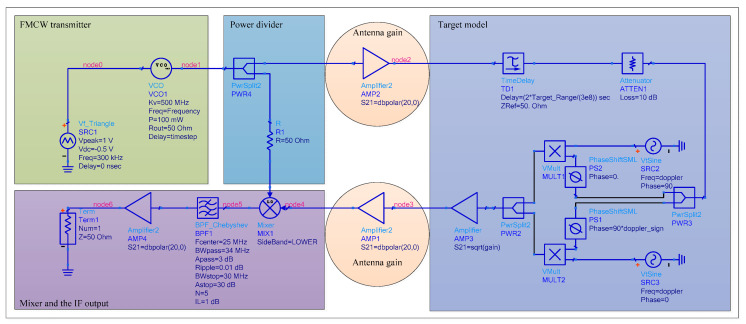
The composition of the circuit model of the FMCW radar.

**Figure 9 sensors-22-04641-f009:**
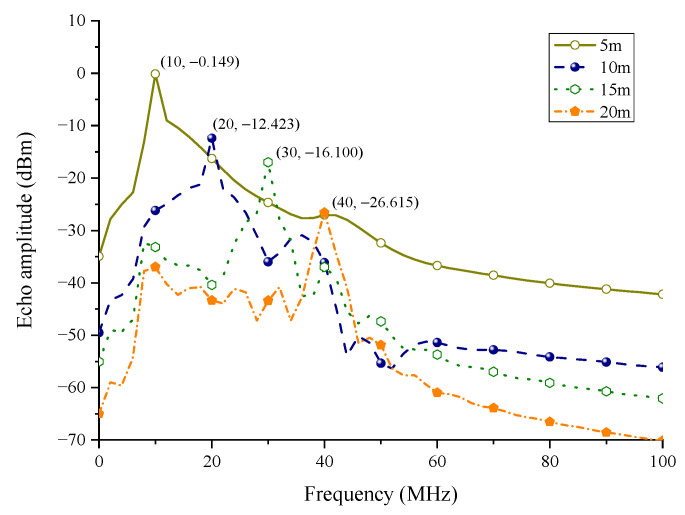
IF output spectrum.

**Figure 10 sensors-22-04641-f010:**
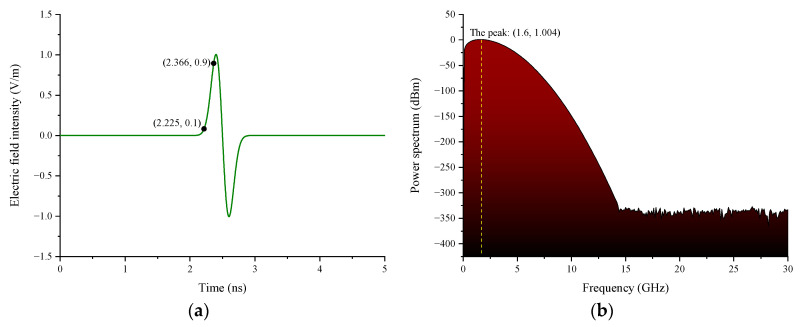
Time-domain waveform and the frequency-domain waveform of UWB EMP. (**a**) Time-domain waveform of UWB EMP; (**b**) frequency-domain waveform of UWB EMP.

**Figure 11 sensors-22-04641-f011:**
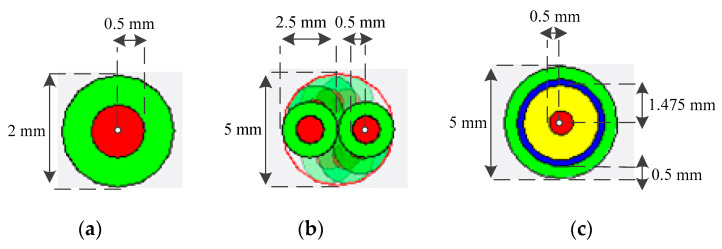
The actual geometry of the wires. (**a**) A single wire; (**b**) A twisted wire; (**c**) A coaxial wire.

**Figure 12 sensors-22-04641-f012:**
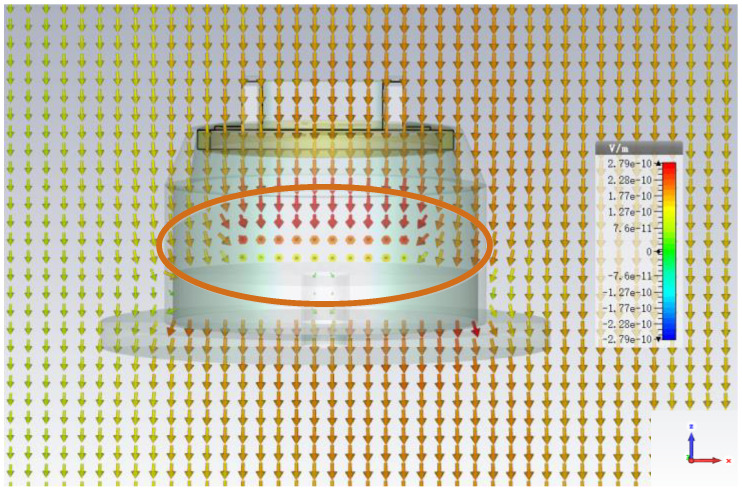
The structure of electromagnetic field.

**Figure 13 sensors-22-04641-f013:**
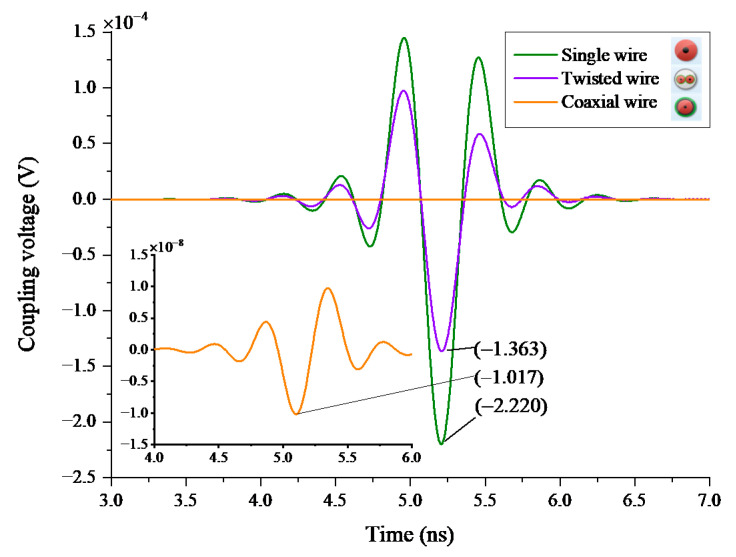
The responses of different categories of metal wire.

**Figure 14 sensors-22-04641-f014:**
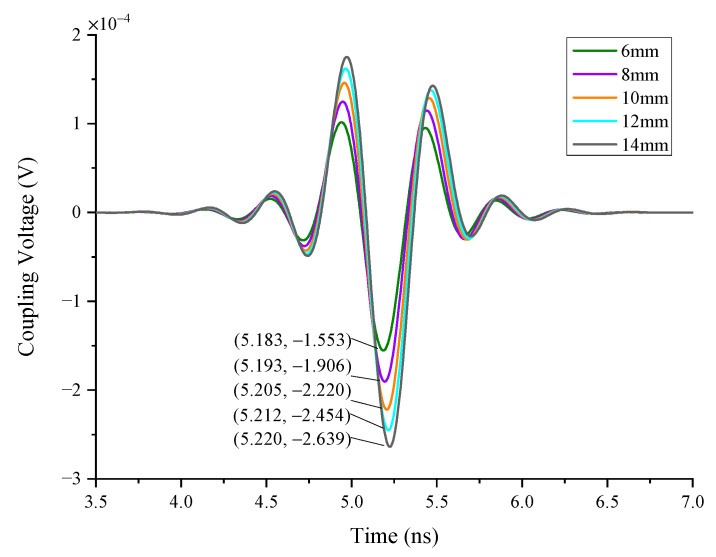
The responses of different lengths of single wire.

**Figure 15 sensors-22-04641-f015:**
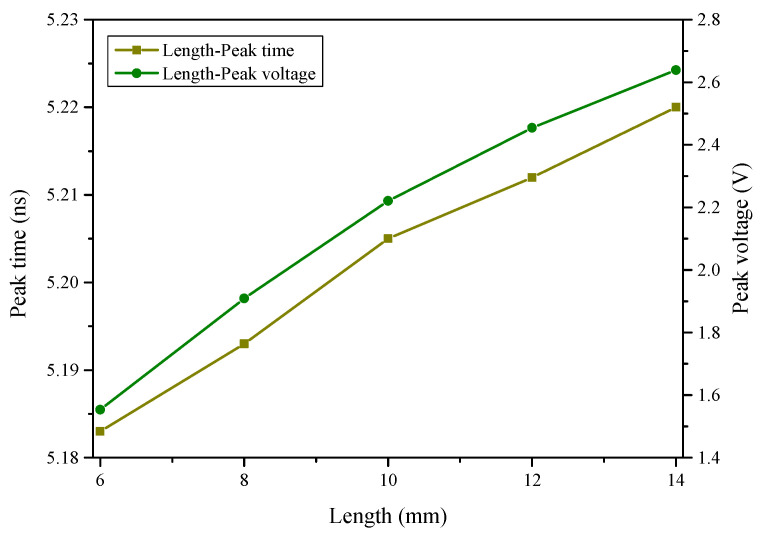
Relationship between peak time and peak voltage with the wire lengths.

**Figure 16 sensors-22-04641-f016:**
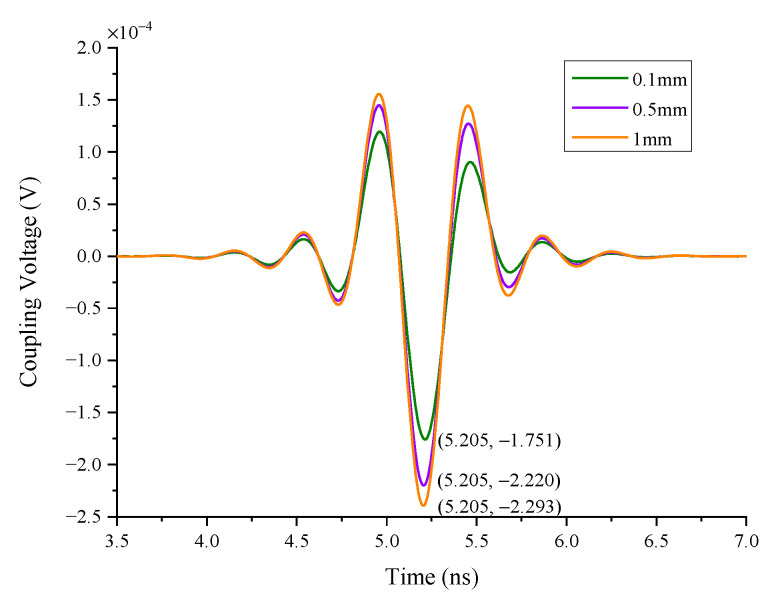
The responses of the different radii of a single wire.

**Figure 17 sensors-22-04641-f017:**
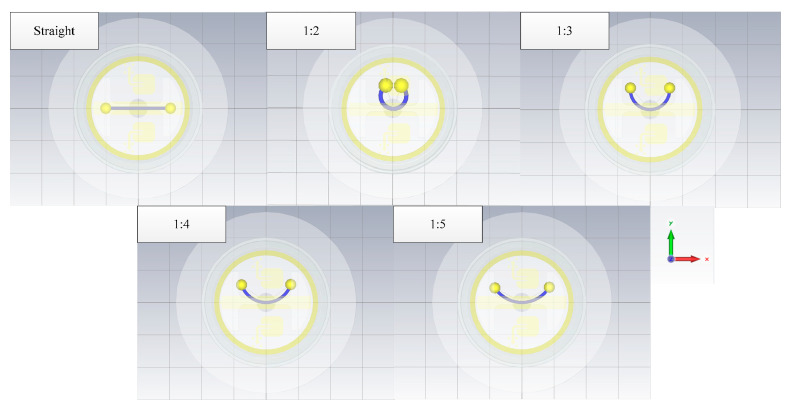
The geometry of the wires.

**Figure 18 sensors-22-04641-f018:**
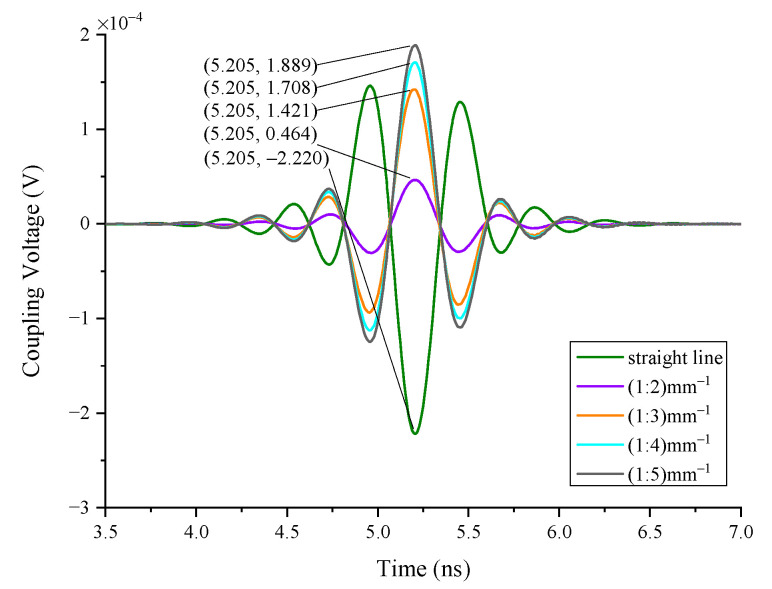
The responses of the different curvatures of a single wire.

**Figure 19 sensors-22-04641-f019:**
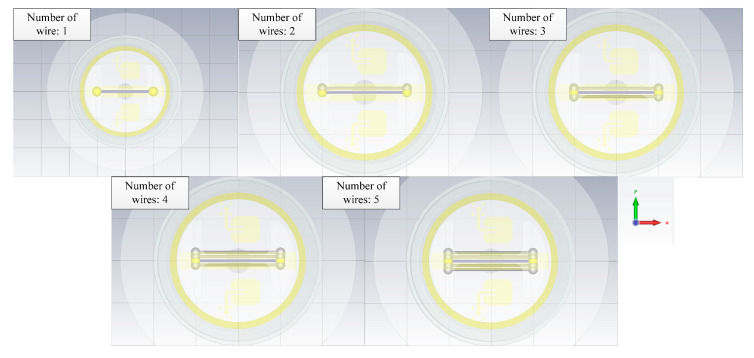
The geometry of the wires.

**Figure 20 sensors-22-04641-f020:**
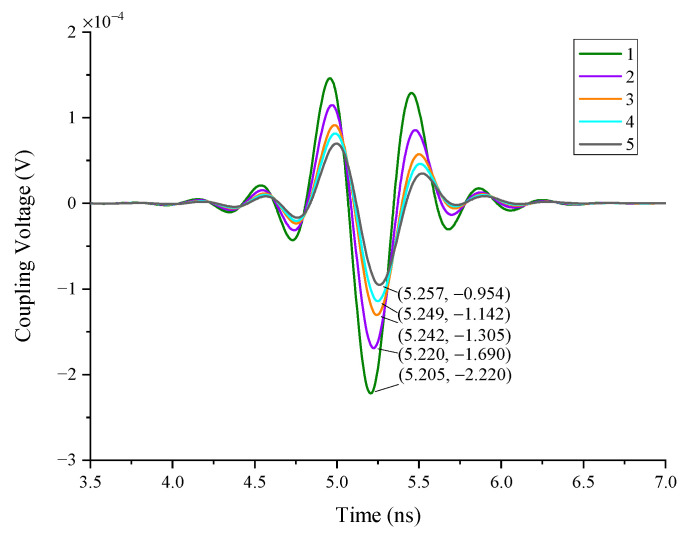
The responses of the different number of wires.

**Figure 21 sensors-22-04641-f021:**
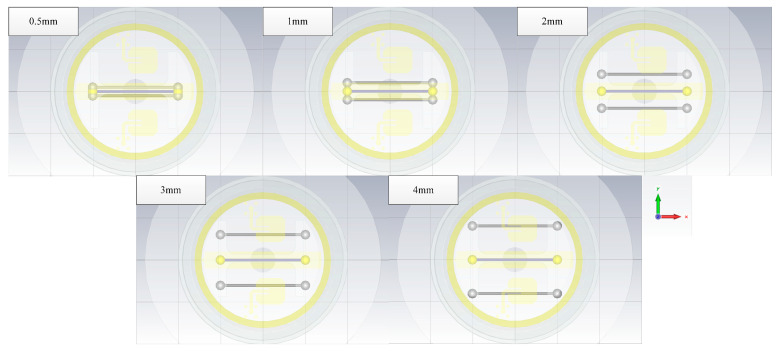
The geometry of the wires.

**Figure 22 sensors-22-04641-f022:**
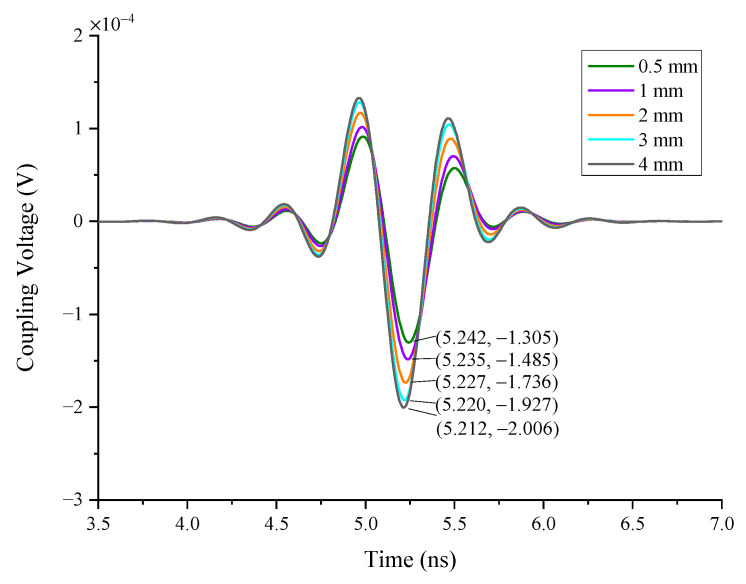
The responses of different distances among the wires.

**Figure 23 sensors-22-04641-f023:**
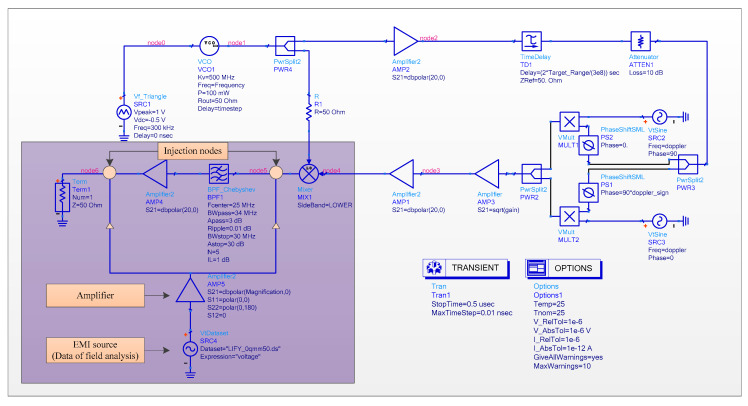
The schematic of the circuit analysis.

**Figure 24 sensors-22-04641-f024:**
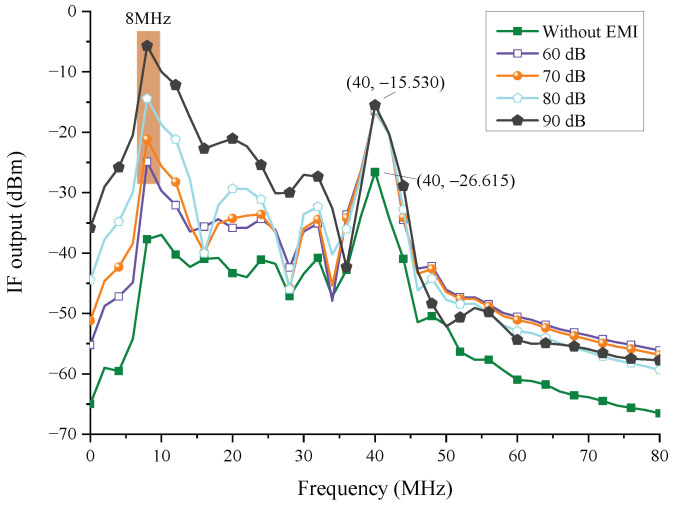
The injection interference law in node 5.

**Figure 25 sensors-22-04641-f025:**
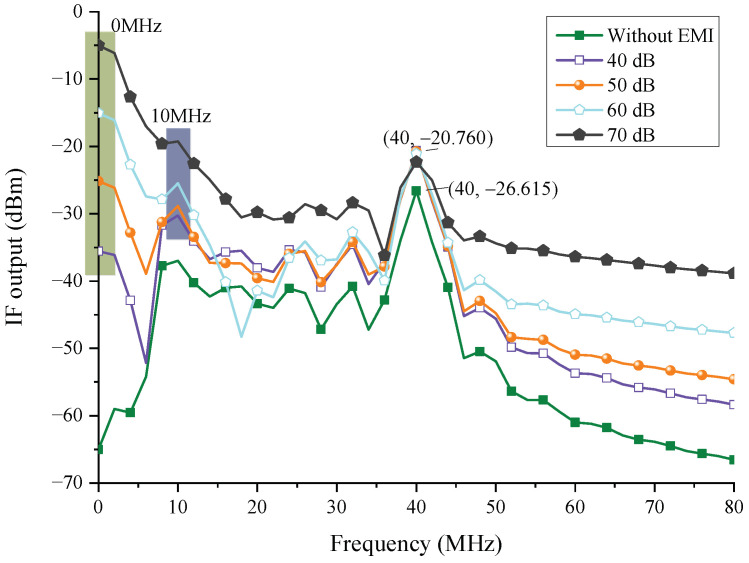
The injection interference law in node 6.

**Figure 26 sensors-22-04641-f026:**
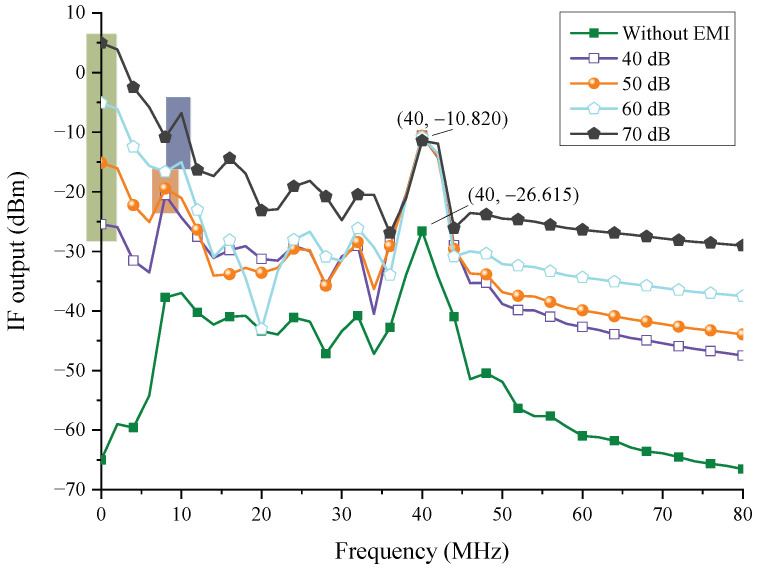
The injection interference law in nodes 5 and 6.

**Table 1 sensors-22-04641-t001:** The comparison of recently reported research results.

References	Methods	Models	Disadvantages
[3,4,5]	Hybrid method	The field-circuit model	Incomplete modeling
[6,7,8,9]	Analytical method	Ideal enclosure	Unable to handle complex enclosures
[12,17]	Numerical method	The actual structure	Unable to handle active circuits
[16]	Experiment method	The actual structure	Expensive

**Table 2 sensors-22-04641-t002:** The parameters of the circuit model.

Transmitter Power	Modulation Frequency	Modulation Bandwidth	Detection Distance	Target RCS	Target Velocity
0.1 W	300 kHz	500 MHz	5–20 m	1 m^2^	30 m/s

**Table 3 sensors-22-04641-t003:** Settings for the simulation cases.

Case No.	Wire Type	Length (mm)	Radius (mm)	Curvature	Number	Line Spacing (mm)
1	single wire/ twisted wire/ coaxial wire	10	0.5	straight	1	/
2	single wire	6/8/10/12/14	0.5	straight	1	/
3	single wire	10	0.1/0.5/1	straight	1	/
4	single wire	10	0.5	straight/ 1:2/1:3/ 1:4/1:5	1	/
5	single wire	10	0.5	straight	1/2/3/4/5	0.5
6	single wire	10	0.5	straight	3	0.5/1/2/3/4

**Table 4 sensors-22-04641-t004:** Other parameters of the wires.

Wire Type	Material of the Conductor	Material of the Insulator Inside	Material of the Insulator Outside	Material of the Screen
LYFY-0 qmm50	Copper	/	Polyvinyl chloride	/
LIFY-1 qmm	Copper	/	Polyethylene	/
RG-58	Copper	Polyethylene	Polyvinyl chloride	Copper

**Table 5 sensors-22-04641-t005:** Minimum amplification of EMI sources.

Target Range/m	A Single Wire/dB			A Coaxial Wire/dB		
	Node 5 Injection	Node 6 Injection	Two-Point Injection	Node 5 Injection	Node 6 Injection	Two-Point Injection
5	105	82	88	189	166	172
10	88	69	75	172	154	159
15	84	64	71	169	148	155
20	78	55	65	162	139	149

**Table 6 sensors-22-04641-t006:** Peak voltage of IF output of a single wire.

Target Range/m	Peak Voltage of IF Output/V		
	Node 5 Injection	Node 6 Injection	Two-Point Injection
5	2.478	5.541	11.500
10	0.469	1.241	2.474
15	0.304	0.697	1.561
20	0.144	0.248	0.782

## Data Availability

All data, models, and code generated or used during the study appear in the submitted article.

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
