# Peer review of "Simulation and Analysis of an FMCW Radar against the UWB EMP Coupling Responses on the Wires"

_sensors, 2022, doi:10.3390/s22124641_

Round 1
Reviewer 1 Report
The article is devoted to the problem of the influence of a ultra-wideband electromagnetic pulse (UWB EMP) on the FMCW radar. The study is based on the two-component model. The field analysis is used to calculate the voltage induced on the elements of the radar's circuit by UWB EMP. The circuit model is used to calculate the effect of induced signals on the operation of the radar. The article describes the specific radar operating in the presence of the specific UWB EMP.
The statement of the problem considered in the article is relevant today and corresponds to the profile of the journal. The article as a whole is clearly structured and easy to understand. The research methodology is relevant to the task and is verifiable. The results obtained are clearly stated. The conclusions are consistent with the results obtained. The cited literature is relevant and mainly consists recent publications. The authors do not resort to excessive self-citations.
There are several points in the article that the authors should clarify in order to make the presented materials more understandable to the reader.
1) Authors should describe in more detail the way in which the results of the field analysis are introduced into the circuit model. For example, in Figure 17, authors should indicate the sources of EMF, simulating the effect of the UMW EMP influence, and their connections with the electrical circuit along with their parameters.
2) Authors should describe the actual geometry of the wires and cables connected to the radar in cases 1-6 described in section 4.1 and the direction of incidence of the UMW EMP. Also, readers might be interested in the structure of electromagnetic fields obtained during the calculation.
Reviewer 3 Report
The paper was easy to follow and all concepts were explained. Some major corrections should be done.
(1) English language and style are fine or minor spell check required.
(2) The authors need to insert a comparison table for recently reported research results.
(3) The authors need to provide novelty of the proposed technique.
(4) The authors need to insert real measurement results.
Round 2
Reviewer 2 Report
The modified content meets the requirements
Reviewer 3 Report
The paper was easy to follow and all concepts were explained. Some minor corrections should be done.
(1) English language and style are fine or minor spell check required.
(2) The authors need to insert a comparison table for recently reported research results.
